# The Role of Physical Therapies in Wound Healing and Assisted Scarring

**DOI:** 10.3390/ijms24087487

**Published:** 2023-04-19

**Authors:** Montserrat Fernández-Guarino, Stefano Bacci, Luis Alfonso Pérez González, Mariano Bermejo-Martínez, Almudena Cecilia-Matilla, Maria Luisa Hernández-Bule

**Affiliations:** 1Dermatology Service, Instituto Ramón y Cajal de Investigación Sanitaria (IRYCIS), Hospital Ramón y Cajal, 28034 Madrid, Spain; 2Research Unit of Histology and Embryology, Department of Biology, University of Florence, 50121 Firenze, Italy; 3Specialist Nursing in Wound Healing, Angiology and Vascular Service, Instituto Ramón y Cajal de Investigación Sanitaria (IRYCIS), Hospital Ramón y Cajal, 28034 Madrid, Spain; 4Diabetic Foot Unit, Angiology and Vascular Service, Instituto Ramón y Cajal de Investigación Sanitaria (IRYCIS), Hospital Ramón y Cajal, 28034 Madrid, Spain; 5Bioelectromagnetic Lab, Instituto Ramón y Cajal de Investigación Sanitaria (IRYCIS), Hospital Ramón y Cajal, 28034 Madrid, Spain

**Keywords:** chronic wound, electromagnetic fields, hypertrophic scar, keloid, laser, physical therapies, photobiomodulation, photodynamic therapy, radiofrequency, ultrasound therapy, wound healing

## Abstract

Wound healing (WH) is a complex multistep process in which a failure could lead to a chronic wound (CW). CW is a major health problem and includes leg venous ulcers, diabetic foot ulcers, and pressure ulcers. CW is difficult to treat and affects vulnerable and pluripathological patients. On the other hand, excessive scarring leads to keloids and hypertrophic scars causing disfiguration and sometimes itchiness and pain. Treatment of WH includes the cleaning and careful handling of injured tissue, early treatment and prevention of infection, and promotion of healing. Treatment of underlying conditions and the use of special dressings promote healing. The patient at risk and risk areas should avoid injury as much as possible. This review aims to summarize the role of physical therapies as complementary treatments in WH and scarring. The article proposes a translational view, opening the opportunity to develop these therapies in an optimal way in clinical management, as many of them are emerging. The role of laser, photobiomodulation, photodynamic therapy, electrical stimulation, ultrasound therapy, and others are highlighted in a practical and comprehensive approach.

## 1. Introduction

Wound healing (WH) is a main health problem in current society. Firstly, acute wounds could lead to scars and disfiguring lesions, and secondly, chronic wounds (CW) cause morbidity and high economic cost. AWs occur, in general, after surgery, trauma, or burns, whereas in CWs occur, in general, with an underlying systemic condition, such as diabetes, elderly, vascular alterations, or malnutrition.

Guidelines for care in wounds are useful, clear, and concise [1]. They represent the principal approach in clinical practice. The main CWs presented in clinical practice include leg ulcers (LU), diabetic foot ulcer (DFU), and pressure ulcer (PU). The main treatment of CWs include adequate dressing, debridement, and pressure control. Nevertheless, undoubtedly, there is an uncovered gap in this pathology, as scarring is sometimes unavoidable, and CWs could persist for months. The focus of this review is to provide a tool for clinicians, and a useful guide from the basic science, to develop and improve physical therapies in WH.

Lots of research works are nowadays focused on solving the problem of WH, most of them searching for very advanced therapies, such as cellular transplantation therapy [2,3], vascular enhancers [4], regenerative materials [5], or nanoparticles in hydrogels [6]. Despite the highly anticipated novel therapies in development, right now, they are very far from being used in real practice.

Physical therapy (PT) is present in daily clinical consultations and has demonstrated a certain utility in WH [7]. This review came across different techniques such as laser, low-level laser light therapy, photodynamic therapy, or electrical stimulation, among others, and their role in WH.

## 2. General Approach to Wounds

### 2.1. Epidemiology

The data in the literature referring to failure of WH show the seriousness of the problem.

WH failure, dermal fibrosis and scarring affect all ethnicities, while keloids or hypertrophic scars are more prevalent in American, African, and Asian populations, which can reach up to 16% of the population [8]. Factors associated with excessive scarring include genetic predisposition, hypertension, endocrine dysfunction, autoimmune diseases, and endocrine alterations [9]. The genetic factors described have been found to be associated with polymorphism alterations in genes such as TGF-beta, which evolved in fibrosis formation, opening, and interested therapeutic target [10]. The subsequent endothelial malfunction in hypertension has recently been associated with the risk of scarring and other diseases which have fibrosis and remodeling in their pathogenic [11].

On the other hand, the type of injury has also been associated with the risk of scarring and other factors often seen in clinical practice [12] (See Table 1). Two types of scars are described: keloids and hypertrophic scars (HS). HS are limited to the wound with an increase in cicatricial tissue, whereas keloids are invasive, going through the limit of the wound [12]. Table 1 summarized the differences between keloids and HS. The interaction between the environment of keloids and the scar is complex, and diet, smoking, stress, and physical exercise could influence the process [11].

Conversely, the failure of healing a wound also produces a high impact on the patients. A chronic wound (CW) is a wound that fails to repair and restore the skin in three months [13]. It is estimated that 1–2% of the population suffer from CWs [14], for example, in the United States more than 6.5 million patients are affected [15].

### 2.2. Process and Stages of Wound Healing

WH is a complex process evolving multiple biological pathways and mechanisms. Classically, it is divided into different phases, including hemostasis/inflammatory stage, proliferation, and remodeling (Figure 1) [2,16].

#### 2.2.1. Hemostasis/Inflammatory Stage

The first response to an injury is the constriction of the affected vessels and platelet activation to form a fibrin clot and stop the bleeding [3]. Platelets are activated for the exposure to collagen of the subendothelial matrix in the so-called primary hemostasis. Next, secondary hemostasis produces the activation of the coagulation cascade [17]. Local mast cell degranulation release occurs in the following minutes, and mediators such as histamine and TNF-alfa are released [18].

The next cells to appear in the wound scenario are the neutrophils which are not usually present on normal skin. Neutrophils represent an innate inflammation [19] and are recruited from damaged vessels and attracted by interleukin 1 (IL1), tumor necrosis factor alfa (TNFα) and bacterial toxins [20].

Activated neutrophils destroy bacteria and cell debris and provide a good environment for WH through the liberation of reactive oxygen species (ROS), antimicrobial peptides and proteolytic enzymes [16]. Clearance of neutrophils occurs by apoptosis or necrosis and ulterior phagocytosis by macrophages. Complete hemostasis and inflammatory phase in WH usually last 72 h [2].

#### 2.2.2. Proliferation

The clot is substituted by connective tissue or granulation tissue, meanwhile neovascularization, re-epithelialization, and immunomodulation appear in parallel lasting days or weeks [16]. Many cytokines and growth-factors participate in this phase, such as the transforming growth factor-beta (TGF-β) family, and vascular epidermal growth-factors (VEGF) [2,16]. Most of these mediators support the mechanism of action of physical therapies in WH and are the focus to work in with. The duration of this phase is as follows: 3–21 days [21].

#### 2.2.3. Maturation/Remodeling

Finally, a progressive substitution of the existing cells in the initial fibrin clot led to a wound contraction. This event is related to the maturation of type I collagen and the elimination of type III immature collagen, and by apoptosis of the myofibroblast during several weeks and months after the injury [22]. This change is regulated by metalloproteases (MMPs), collagenases express and secrete by macrophages, myofibroblasts, and keratinocytes [3,16]. The duration of this phase is as follows: 3 weeks–6 months [21].

### 2.3. Chronic Wounds

A chronic wound (CW) is described as a wound that fails to repair itself or remains unhealed after 12 weeks [1].

Most of the CW are classified as diabetic ulcers (DU), pressure ulcers (PU), or venous leg ulcers (VU), in relation to their clinical findings and cause.

#### 2.3.1. Diabetic Ulcer

DU is a deleterious complication representing the first cause of amputation of the lower limb [23]. DU are located on the foot and are caused by neuropathy and vascular illness, which causes the inability to detect pain and injuries. In general, DUs are deep, similar to a crater and expose the tendon and the bone. A surrounded hyperkeratotic tissue is put in place, forming a callus-like ring. Imaging testing could be necessary to exclude osteomyelitis [1,13].

#### 2.3.2. Pressure Ulcer

PU appears on areas under pressure, usually over a bony prominence such as the sacrum or the heels. The pression on the vessels decreases irrigation of the skin resulting in an initial dermatitis, which if prolonged leads to a loss of tissue. The cause is multifactorial, e.g., immobilization in bed, nutrition alteration, systemic diseases, or being elderly [1]. PUs varies in severity, and are classified in four different stages, IV being the more severe, which implies the loss of the full thickness of the skin. In those cases, the management of the PU should be surgical, and PT would not play a role.

#### 2.3.3. Venous Ulcer

A VU typically appears in the lower limb over the medial supramalleolar area. The risk factors for VUs include obesity and venous insufficiency. About 75% of chronic ulcers are VU, being the most frequent, affecting 1–5% of the population [24]. VU are associated with more signs of venous malfunction in clinical exploration than oedema, hemosiderosis, cutaneous atrophy, lipodermatosclerosis or annexal absence. If necessary, further examination with a duplex ultrasonography confirms the diagnosis.

The management of a VU includes the general treatment of CW, adding compressive therapy and healing the venous system if possible, with surgery [13,25]. Adjuvant therapies include nutritional balance and supplementation, diet, physical exercise and improving blood circulation with drugs such as pentoxifylline. Despite using the correct treatment, a VU could take 6 to 12 months to heal, and relapsing is very frequent in the following year [13,24,25].

### 2.4. General Management of Chronic Wounds

WH and scarring is a complex process with multiple influencing and interacting factors. Additionally, some of those factors are not under the control of the dermatologist, such as age, vascular abnormalities, comorbidities, malnutrition, or smoking [1]. The management is challenging, and multiple approaches and visits are needed with the implication of different health care workers [1] and arisen important indirect costs.

All CW should be treated according to the TIME principles: tissue debridement, infection control, moisture balance and edges of the wound [13]. Debridement is the first step in the treatment of a CW, it must be carried out weekly and it increases the speed of healing by over 72% [26].

Biofilm is presented in the extracellular matrix and is considered the cause of 80% of the infections in CW [27]. Biofilm is invisible to the naked eye, and different techniques to assess its presence are being developed, apart from a cutaneous biopsy. Nevertheless, the biofilm must be removed because it maintains the CW in the inflammatory stage [28]. The risk of infection is usually controlled by topical antibiotics, silver dressing, or with other topical components.

The wound should not be exposed to air, and if the skin appears dry, moisturizer should be added to the dressing. On the other hand, if excessive drainage is present, it needs to be clean and dried. The wound edge, in case of overgrowth, must be excised for epithelization [29].

Table 2 describes the local cellular response alterations underlying a CW. CW are characterized by excessive inflammation, a decrease in growth factors secretion, and a disbalance in the proteolytic enzymes and cellular senescence which perpetuates the wound unhealed [25]. Therefore, a high number of mast cells, neutrophils and dendritic cells are found in CW with an increase in pro-inflammatory cytokines and proteases (see Figure 1). These inflammatory cells not only prolong the wound but also increase susceptibility to infections [22]. The alteration of the expression and activation of MMPs is strongly associated with CW, damaging the granulation tissue, and producing exudates in the wound [30]. Cells implicated in the remodeling and re-epithelization are dysfunctional too. Fibroblasts are senescent and the excess of wound proteases (MMPs, elastase, cathepsin G, and urokinase-type plasminogen activator (uPA)) activities degrade the extracellular matrix, the growth factors (VEGF, TGF-beta) and cytokines (TNF-alfa) [22]. Keratinocytes hyper proliferate at the edge of the wound, so hyperkeratosis appears, and the subsequent wound fails to close. The microbiome profiles of aged and diabetic patients with CW have been found to have a decrease in alfa-diversity [3,20,30].

### 2.5. General Prevention of Scarring

Excess WH or scarring is caused by an overproduction of extracellular matrix generated by myofibroblasts, which in this type of lesion are not replaced by fibroblasts during the proliferative phase. In this fibrosis, matrix proteins such as alpha-smooth muscle actin (alpha-SMA) are overexpressed, and the expression of MMP decreases, which induces an accumulation of collagen [12].

Keloid and HS are clinical expressions, and both can be considered successive stages of the same proliferative disorder. The initial common process is a purulent inflammatory skin lesion, the hyperfunction of the fibroblasts and excessive extracellular matrix deposition. HS consists of mainly type III collagen, whereas keloids contain type I and III [31].

The general strategy for the prevention of scarring is summarized in Table 3. The early recognition of the alteration is considered of cardinal importance for early treatment [32]. The healing process varies from one patient to another; thus, controlling the procedure, preventing the infection, and providing personalized wound care are the main prevention and treatment methods [33].

## 3. The Role of Physical Therapies in Hard-To-Heal Chronic Wounds

### Principles and Basis

Once it is known what fails in WH, the possibility of understanding the role of physical therapies arises more clearly. Figure 2 shows a scheme of possible targets for increasing WH, and Figure 3 shows how to promote regeneration rather than scarring with physical therapies (PT). It is of notice that with their theoretical mechanisms, we can impact in all the phases completing and fostering traditional treatments with innocuity.

The guidelines for the management of CW are extensive, but the pillars are promoting patient adherence to treatment, debridement control of the possible infection, covering with an appropriate dressing and effective compression if necessary [1].

Two options appear when PT are introduced in the treatment, either CW or scarring. One is proactive management, starting the treatment in the initial phases of the wound, as a prevention or adjuvant therapy. The other one is using those therapies when a CW, HS or keloid has appeared. Both situations not only depend on the physician but also the patient consultation.

## 4. Physical Therapies in Assisted Healing and Scar Prevention

### 4.1. Laser

The main target of laser therapy is the treatment and prevention of scarring, and there are few studies published in its assistance in WH. Among the different issues presented in an HS or keloids, different lasers could be used to target each objective [34] (Table 4). Laser treatment is flexible and allows for their combination in a single treatment session. The most widely applied are fractional lasers in combination with vascular lasers and lasers targeting melanin [35]. Basically, there are two different types of fractional laser: ablative (wavelengths of 2790–10,600 nm) and non-ablative (1320–1927 nm). Both ablative and non-ablative lasers have become the gold standard for the treatment of scarring, although ablative lasers are probably the most used [36].

Erbio and CO_2_ lasers are ablative lasers that target water, producing a selective burn in the skin. In a fractional mode, they work in separated columns, allowing for a better regeneration throughout the non-damage columns of skin. Both induce selective thermal necrosis in the skin, increasing in the first weeks of the inflammatory stage of the scar, but after three months, collagen remodeling in a thin bundle due to collagen III appears [37]. The clinical results show a decreasing dermis thickness and increasing skin flexibility [35]. On the other hand, vascular lasers target small vessels and are used for decreasing erythema in HS and keloids, causing excessive neovascularization. Pulsed dye laser (PDL) is probably the most used. PDL has been demonstrated to decrease connective tissue growth factor expression in keloids, despite targeting vessels [38] and inhibiting fibroblast proliferation in vitro [39]. After the vessel coagulation and subsequent hypoxia, PDL leads to an increase in collagen type III [40].

Apart from the treatment of scars, some studies of lasers in assisted WH and preventing scarring from the first day of surgery have been published showing different results. Curiously, the immediate application of lasers differs from other physical therapies, which need some healing days before they start to be applied. In a split-face study, no differences were found in the area treated with CO_2_ laser immediately after surgery, but in other similar studies the scars treated exhibit better healing and cosmetic outcome [41,42,43]. PDL and non-ablative fractional laser have also been shown to improve scaring when used early; however, the differences with the untreated area were not statistically significant [44]. Different types of lasers could be applied in the same session, PDL plus ablative fractional CO_2_ laser have been suggested to be the best combination [45].

An early start of the treatment is recommended in the literature reviewed when lasers are used to assist scarring. The optimal interval between sessions has been found to be 5 to 6 weeks during a period of months [46]. All the lasers applied in early treatment times have also been used under lower parameters [40]. Further clinical trials with long-term follow-up are needed to support the evidence of laser treatment in HS, keloids and WH alone or in combination with other options for treatment [47]; however, lasers are recommended for expert panels as a first-line therapy in scarring [48].

### 4.2. Photobiomodulation (Low-Level Light Therapy-LLLT)

LLLT has been intensely studied in WH, the near-infrared light (NIR) between 800 and 900 nm and red light (600–700 nm) being the most used. The use of light in a non-thermal effect is supported by the photon’s absorption of the cells’ receptors. The main three chromophores in the skin are melanin in the epidermis, hemoglobin in the dermis and water in all the skin [49] and longer wavelengths achieve deeper penetration (Figure 4).

Hormesis responses occur in WH in response to low doses of light (LLLT) or photobiomodulation (PBM). Hormesis or biomodulation are terms used to describe a natural biological process in which low doses of an input, for example, light or energy, induce activation, but high doses produce an inhibition [50]. PBM induces the production of nitric oxide (NO), a vasodilator, and anti-inflammatory agent (Table 5) [51]. LLLT can trigger natural mechanisms involved in WH, including TGF-beta families of molecules, transforming growth platelet factor, interleukins (IL6, 13, 15), and matrix metalloproteinases (MMPs) associated with alterations in WH. TGF-beta is crucial in fibroblast proliferation [7,50,51]. Thus, PBM has been demonstrated to be useful in all the steps of WH.

In animal models, LLLT increases collagen and reduces oxidative and nitroxidative stress in diabetic wounded mouse skin [51]. In vitro studies have also found an increased expression in keratinocytes after LLLT of cyclin D1 and cytokeratin, suggesting an increase in proliferation and maturation [52,53].

LLLT is not as widely used as laser despite being safer, without adverse reactions such as swelling, crusting or purpura. With respect to laser, LLLT is easy to apply, allows the treatment of bigger areas, a wearable device is available, self-treating is an opportunity and it is not as expensive. The main disadvantage of LLLT is the necessity of near-daily repeated sessions [54].

There are few studies of LLLT in WH with different results. In VU, red light did not demonstrate any additional benefit to conventional treatment [55]. Whereas in PU and DU, red light increases healing with better outcomes when compared with NIR [56]. A prophylactic treatment in the prevention of keloid in three patients was shown effective with NIR (LED 805 nm). In this small study, patients self-treated at home daily for one month [57].

LLLT improves inflammation, releases pain, and fosters healing in clinical practice. Even though it has been deeply investigated, further studies in the daily clinical application are necessary as no standard protocol has been developed [54].

#### Blue Light Emission Diode

LED technology greatly benefited from the pioneering research conducted during the gallium nitride crystal boom of the 1980s by Akasaki, Amano, and Nakamura, which led to the invention of the blue LED. This discovery was extremely important as it made it possible to obtain white light from LED sources, paving the way for revolutionary uses of radiation [58].

Blue-light PBM triggers a cascade of events attributable to the absorption of photons by intracellular photoreceptors. Among these effects, the impact of light on cytochrome-C oxidase can be observed: it induces an increase in cell proliferation, migration and differentiation, cytokine modulation, growth factor synthesis, and anti-inflammatory effects; thus, stimulating the improvement of the healing process [59,60].

In wounds treated with blue light, a faster healing process and better deposition and morphology of dermal collagen are observed when compared to wounds not treated with blue light. Furthermore, treated wounds show better modulation of the inflammatory response where mast cells assume a central role [50].

### 4.3. Photodynamic Therapy

Photodynamic therapy (PDT) is a safe and easy procedure to enhance WH, nevertheless, further studies are necessary to determine an exact protocol. Anyhow, PDT is versatile, with the limitation of pain during the treatment and repeat sessions.

PDT is indicated in dermatology for the treatment of actinic keratosis, basal cell carcinoma and Bowen disease [61]. PDT has been explored in WH and prevents scarring, whereas no results have been found in the treatment of keloids and HS. The main difference between PDT and other PT in WH is the ability to scope infections without resistance to antibiotics.

PDT consists of the combination of a photosensitizer (PS) in the target tissue and the subsequent illumination of an adequate light source for inducing necrosis and apoptosis of the tissue. Through the literature, a variety of lights and PS have been tested in WH. Nowadays, PS are preferred to be used topically, as they have lower side effects. A lot of optimal light sources could be used in the PpIX absorption spectrum; however, LEDs are mostly explored for their simplicity and lower side effects. Table 6 summarized which PS could be used in WH and different light devices [62,63]. Most of the light sources are in the red spectrum [63], although there are studies with green light. The protocols and doses for the use of PDT in WH are very different, which is a limitation when trying to come up with a conclusion [64].

The PS increases the intracellular production of Protoporphyrin IX, which absorbs the light and produces the reaction. Destruction is mediated by the production of excessive intracellular ROS (radical oxygen singlet).

The mechanisms of action of PDT are well known; besides the necrosis of the tumors, a lot of parallel biological phenomena are produced, which lead to exploring other indications of WH (Table 7) [65].

PDT produces the activation of acute inflammation in WH, fostering the natural process, and consequently, the neutrophils, TNF-alfa, and IL6 become increased [66]. PDT also induces neovascularization induced by VEGF needed for remodeling [30].

Additionally, studies have indicated that the early activation of fibroblasts and re-epithelization and increase in degranulation index by mast cells play a crucial role in the healing of chronic wounds. It is worth remembering how interactions of the immune system with the nervous system are important in the regulation of wound healing processes. Recent studies have demonstrated that MC interactions with neuronal cells containing neurotransmitters involved in wound healing processes, such as CGRP, NGF, NKA, NPY, SP, PGP 9.5, and VIP, are common in chronic wounds. This fact can be related to other facts such as the secretion of extracellular matrix by fibroblasts, as well as increases in TGF beta levels and the response of cellular infiltrates [18,30].

Afterwards, PDT a negative regulation of the inflammation appears with IL10 expression and down regulation of IL1 and IL6 [67]. It has been suggested that the modulatory effect of PDT in the immune system and the necrosis versus apoptosis induction depends on the intensity of the protocol [68], more specifically on the ROS levels. Therefore, high intracellular production of ROS could change the activation into destruction (hormesis) [69].

PDT increases the levels of MMPs after three weeks, and the histological improvement appears at nine months. On the other hand, PDT has antibacterial activity, targeting the biofilm, which is responsive to chronic inflammation [30,70].

### 4.4. Electrical Stimulation

Endogenous bioelectric fields (EBF) take place during WH, produced by the cells generated by the Na^+^/K^+^ ATPase of the epidermis. EBFs influence cell migration, proliferation, and function, but also gene and protein expression [71]. The underlying mechanisms presented in a CW could be targeted with electrical stimulation (ES) mimicking the natural process (Figure 2). Table 8 summarized the effects of ES in WH outlining in which part of the process this mechanism is working. Theoretically, ES offers benefits in WH after some days in the wound improving the proliferation and remodeling phase. Moreover, if a CW is established, ES could decrease inflammation and the risk of infection. In vitro studies have demonstrated a decrease in *Staphylococcus aureus*, *Pseudomonas aeruginosa* and *Escherichia coli* with ES [72]. Positive scattered results with ES have been reported in CW, VU, and LU, with a possible increase of 30–42% in the reduction in the wound area [72,73]. ES is a safe, simple, cheap, and easy procedure to use without adverse effects.

There are different forms of ES, including direct current, alternating current and pulsed current on mono or bipolar devices. That huge variability limits knowing the real exact beneficial protocol. Moreover, no comparative study between those modalities has been conducted, whereas it is supposed that the pulsed current is the most similar to the physiological(25) [25,71,72]. Theoretically, not all forms of ES are beneficial in all phases of WH, the alternative current only being useful in the first days [25]. There is also a lack of literature about standard protocols [73].

According to the mechanism of action of ES, it would be more effective in the proliferative and remodeling phase of WH, that implies, from days to months after the injury, either in acute wounds (AW) to prevent scarring or in CW to enhance healing [73].

If ES is applied, it should be added to the conventional treatment of the wound as a complementary treatment (Table 9). The ES devices are usually applied by setting electrodes around the wound. Repeated-weekly sessions are necessarily, lasting from 45 min to hours [25]. The therapy could last months, which is a great limitation due to time consumption and displacement. Therefore, ES might be used in selected patients with a risk of failure in WH. Novel devices are emerging, offering different possibilities such as home devices, electric dressings, or electric fields, providing a practical future [71].

### 4.5. Others

#### 4.5.1. Ultrasound Therapy

Ultrasound therapy (UT) consists of sound waves that cause thermal and non-thermal effects in tissues. When UT is strongly applied to the skin, the temperature will rise to 40 Celsius degrees and produces an increase in vessel flow, cell proliferation, collagen synthesis, and tissue regeneration. Moreover, UT has anti-inflammatory properties. The non-thermal effects comply with acoustic streaming with a displacement of the particles and cavitation with the generation of microenvironmental gases [74]. Cavitation cleans necrotic tissue preserving the healthy one [7].

UT accelerates the decrease in the wound area with respect to controls in LU, and it is approved as an adjuvant therapy in WH by the FDA [75,76].

Two types of therapeutic US exits are low frequency ultrasound (LFU) from 30 to 40 kHz and high-frequency ultrasound (HFU), ranging from 1 to 3 MHz. HFU has been used for decades for the treatment of muscular diseases in sports medicine. A variant of HFU, micro focused ultrasound in high intense mode (MFU, HIFU), is being widely studied because of its benefits in aesthetic medicine reducing wrinkles and laxity of the skin [75]. In contrast, LFU has demonstrated efficacy in WH and has been applied with good results in LU.

LFU is used directly on the skin, around the wound for 5 to 10 min (Table 10). A topical gel is usually needed between the skin surface and the applicator [74]. UT is contraindicated in a patient carrying a metal prosthesis in the leg, neuropathy, infection, or thrombophlebitis [75].

UT has a possible application in WH; nevertheless, there are no clinical studies of the effects of UT in WH or scar prevention, most of the evidence is limited to LU and further randomized clinical essays and protocols are necessary [75,76].

#### 4.5.2. Electromagnetic Fields

Low frequency pulsed electromagnetic fields (PEMF) can accelerate WH, generating connective tissue, enhancing the VEGF pathway and the production of collagen type I. There are some published studies with good results in PU, VLU and DU. PEMF are possible to apply at home on portable devices as multiple sessions are necessary [77,78].

#### 4.5.3. Biophotonic Therapy

Biophotonic therapy (BT) consists of the application of the PBM applying a special gel over the CW containing the chromophores. Afterwards, an LED lamp with a hyper pulsed beam and low energy is used to activate the photoconverter gel. One concrete device known as “Lumihel^®^” was evaluated showing improvement in the healing of the CW, increasing the life quality of the patient without adverse events. The main limitations of the study were the simultaneous inclusion of VU, LU and PU and the weekly treatment sessions lasting 8 weeks [79].

#### 4.5.4. Visible Polarized Light

Visible polarized light (VPL) has been used as a complementary therapy in WH. The device used emitted light like the sun but without ultraviolet radiation. Thus, the light used was safe, low energy light, polychromatic, incoherent, and polarized. The polarization allows it to work on flat surfaces and enhances light penetration. The molecular mechanism of action of VL is not well documented; however, some studies showing improvement in treated CW have been published [80,81]. PVL seems a promising possible treatment for WH but needs to be more deeply studied [82].

#### 4.5.5. Radiofrequency

Radiofrequency therapies consist of the application of a high-frequency electromagnetic field (3 kHz and 300 GHz) that induces oscillation and friction in the molecules of the target tissue, which causes tissue hyperthermia. This electrically induced hyperthermia can degrade collagen, which stimulates neocollagenogenesis and tissue remodeling [83]. The main indicators of RF are skin tightening, the reduction of wrinkles and the treatment of scars. There are some studies assessing the efficacy of RF in WH with good results in releasing pain. Nevertheless, multiple sessions are necessary for 2–4 weeks. RF technology is rapidly developing, with new micro-needling devices and fractionated delivery, which shows good results in acne scars, HS, and keloids [21,84].

## 5. Summary of Clinical Trials of Physical Therapies in Wound Healing

Finally, it is important to take into consideration that there are few clinical trials published about interventions with PT in WH. When the term “Physical therapies skin wounds” is introduced in “Pubmed” and filters added as, “last 10 years”, “clinical trials” a total of 66 results appeared. After having been selected, only 8 works were focused on skin and only 6 in CW (Appendix A).

PT are not included in clinical Guidelines of management of CW, and they are used as adjuvant therapies [1]. Polak treated with electrical stimulation (ES) areas of pressure injuries in patients with neurological damage. A total of 43 patients were divided into three groups, anodal, cathodal and placebo. A diminution of proinflammatory blood cytokines was found in the patients treated in correlation with an improvement in the clinical peripheric inflammation of the wound and a significant reduction in the size of the wound was assessed, in comparison to the placebo group. The same therapy was used for other authors [85] to explore the effect in CW in combination with silver dressing. Ten patients were treated only in one wound and the rest were used as controls, leading to significant differences. ES has also been proved to alleviate pain in CW, in addition to accelerating healing [86]. In a study of 10 patients treated with ES compared with 10 patients treated with placebo, ES improved in DU the blood levels of VEGF and NO [87].

In an interesting study, ultrasound (US) and electrical stimulation (ES) were compared in the treatment of PU. Both treatments improved WH in 27 patients treated without significant differences between them [88]. Another comparative study was carried out by Polak in 77 patients with PU using standard wound care, US, and ES. Patients treated with PT had a significant decrease in the ulcer area compared with placebo without differences between US and ES [89]. The main limitation of all those clinical trials is control over others factor that could influence the healing of the wound.

## 6. Conclusions and Future Perspectives

WH and pathological scars are important problems in daily practice causing pain and morbidity and are difficult to manage. PT arises as a possible safe complementary treatment that might improve the results of the traditional treatment. PT has been demonstrated to improve tissue healing with different grades or evidence; however, further studies are necessary to develop practical protocols in clinical practice based on the theoretical mechanism of action of the therapy. The main limitations of PT are the lack of clinical trials, the availability, the variability of the parameters used in different conditions and the lack of comparable results. Probably, electrical stimulation and ultrasound are the most studied. The scope of this review was to offer a complete view of PT for clinicians in WH so they can start working up new adjuvant protocols.

## Figures and Tables

**Figure 1 ijms-24-07487-f001:**
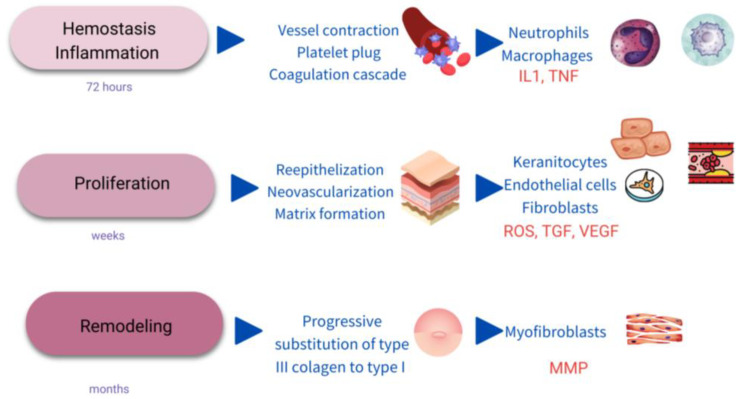
Scheme of the stages of wound healing. IL1: interkeukin1; TNF: Tumor necrosis factor; ROS: single oxygen radicals; TGF: transforming growth factor; VEGF: vascular endothelial growth factor; MMP: metalloproteinases.

**Figure 2 ijms-24-07487-f002:**
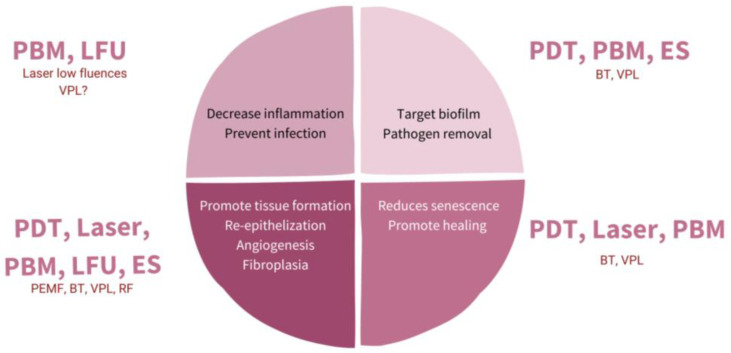
Diagram representing different targets with physical therapies (PT) for promoting wound healing (WH). PBM: photobiomodulation; LFU: low frequent ultrasound; PDT: photodynamic therapy; ES: electrostimulation; VPL: visible pulsed light; PEMF: pulsed electromagnetic fields; BT: biophotonic therapy; RF: radiofrequency.

**Figure 3 ijms-24-07487-f003:**
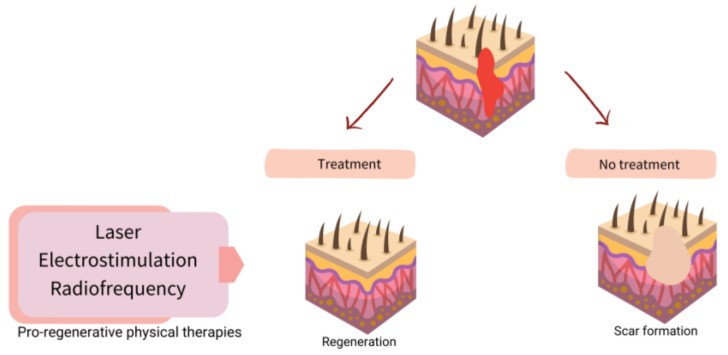
Scheme of strategies for physical therapies (PT) in assisted well-scaring.

**Figure 4 ijms-24-07487-f004:**
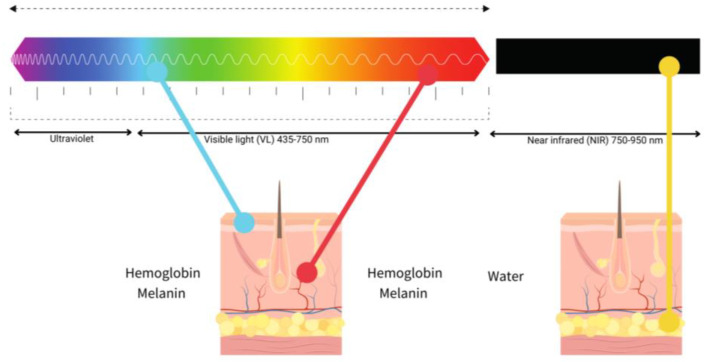
Diagram of the relationship between visible light (blue, red) and near-infrared (NIR), penetration and chromophores.

**Table 1 ijms-24-07487-t001:** Clinical differences between keloids and hypertrophic scars.

Characteristic	Keloids	Hypertrophic Scars
Trauma	Non-severe (acne, folliculitis)	Burns, incision
Body sites	Chest, upper back, earlobe	Any
Symptoms	Erythema, itch, pain	Erythema, itch
Exploration	Beyond the limit of the trauma	Limited to the initial wound
Treatment	Combined therapies with frequent recurrence	Good response
Surgical excision	Contraindicated due to recurrence	Without recurrences, could be considered a treatment

**Table 2 ijms-24-07487-t002:** Mechanisms found in failure to heal wound (FHW).

Wound	Cellular Mechanisms	Mediators
Inflammation Exudates Infection	Neutrophils’ excessive number and function Defective macrophages High number of mast cells Loss of microbiome diversity	Oxidative stress Wound proteases (MMPs, elastase, cathepsin G, and urokinase-type plasminogen activator (uPA)) Increase in inflammatory cytokines
Hyperkeratotic edge of the wound	Keratinocyte hyperproliferation and malfunction	Elevated b-catenin and c-myc
Failure to heal and close	Senescent fibroblasts	Degradation of VEGF, TGF-beta, and TNF-alfa

MMP: metalloproteinases; VEGF: vascular endothelial growth factor; TGF: transforming growth factor; TNF: tumoral necrosis factor.

**Table 3 ijms-24-07487-t003:** Prevention and treatment of scarring, keloids, and hypertrophic scars (HS).

Prevention	Treatment	Alternative Therapies
Early diagnosis	Silicone gel or dressing Topical retinoids	Peelings
Careful wound care	Topical Imiquimod Topical 5-Fluorouracil Intralesional Bleomicin	Microneedling Dermabrasion Radiotherapy
Prevent infection		
Sun protection Avoid risk areas if possible Avoid risk patients if possible		

**Table 4 ijms-24-07487-t004:** Targeting lasers according to clinical exploration in hypertrophic scars and keloids.

Skin Alteration	Type of Laser
Erythema	Pulsed dye laser Intense pulsed light Neodimio-Yag laser
Skin thickness/Hypertrophic	Erbio laser CO_2_ laser Non-ablative laser
Hyperpigmentation	Alejandrita laser Intense pulsed light KTP 532 nm Q-switched laser

**Table 5 ijms-24-07487-t005:** Summary of the beneficial effects of photobiomodulation (PBM) in wound healing (WH) and chronic wound (CW).

Effect	Mediator	Phase of WH/CW
Anti-inflammatory	ROS, NO, IL	Inflammation
Vasodilatation	NO	Proliferation
Matrix formation	TGF-beta and MMPs	Proliferation and remodeling
Promote epithelial cell function	Cyclin D1	Proliferation and remodeling

ROS: radical oxygen singlet; NO: nictric oxide; IL: interleukins; MMPs: metaloproteinases.

**Table 6 ijms-24-07487-t006:** Summary of some of the photosensitizers (PS) and light sources used in photodynamic therapy (PDT) in wound healing (WH).

Group of Photosensitizers	Molecule	Light Sources (570–800 nm)	Protocol
Hematoporphyrin derivates	Photofrin^®^	LED (red and NIR)	37–100 J/cm^2^/session
PpIX precursors	ALA, MAL	Laser (Vascular and Diode)	
Clorins	Foscan^®^ (mTHPC)		1–2 sessions/week/1 month

PpIX: Protoporphyrin IX; ALA: aminolaevulinic acid; MAL: Methylaminolaevulinic acid; mTHPC: meso-tetra-hydroyphenyl chlorin; LED: ligh emitting diodes; NIR: near infrared.

**Table 7 ijms-24-07487-t007:** Summary of the mechanism of action revised of PDT in WH.

Effect	Mediator	Phase of Wound Healing
Activation/suppression of the immune system	TNF-alfa IL1, IL6, IL10	Inflammation
Antibacterial activity	ROS	Chronic inflammation
Reepithelization, matrix formation	MMPs	Regeneration and remodeling
Neovascularization	VEGF	Regeneration and remodeling

TNF: tumoral necrosis factor; IL: interleukins; ROS: radical oxygen singlet; MMPs: metalloproteinases; VEGF: vascular endothelial growth factor.

**Table 8 ijms-24-07487-t008:** Beneficial effects reported of electrical stimulation (ES) in wound healing (WH) and chronic wound (CW).

Effect	Mediator	Phase of WH/CW
Angiogenesis	VEGF	Proliferation
Fibroblast proliferation	FGF	Proliferation and remodeling
Reduces bacterial colonization	PH alteration	Persistent inflammation and risk of infection

EGF: vascular endothelial growth factor; FGF: fibroblast growth factor.

**Table 9 ijms-24-07487-t009:** Summary of the practical initial application of electrical stimulation (ES) in wound healing (WH).

Modality	Type of Wound	Not yet Studied/Not Beneficial
Pulsed current Electrodes around the wound From 30 min to hours		Acute current in CW
From 5 to 7 days a week	DU	Scarring prevention or treatment
After 2–5 days of the injury	LU	

DU: diabetic ulcer; LU: Leg ulcer.

**Table 10 ijms-24-07487-t010:** Summary of the practical protocols reviewed in ultrasound therapy (UT) in wound healing (WH).

Modality	Type of Wound	Not yet Studied/Not Beneficial
LFU Around the wound	LU	HFU Contraindicated metal prosthesis, infection, or neuropathy
From 5 to 10 min Repeated sessions in a week		Prevent or treat keloids or HS
After 2–5 days of the injury		

LU: Leg ulcer; HFU: high-frequency ultrasound; HS: Hypertrophic scars.

## Data Availability

All the data reviewed in the manuscript are published in the articles mentioned in the references and selected for being writing in English and indexed in PubMed.

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
