# Peer review of "The Role of Physical Therapies in Wound Healing and Assisted Scarring"

_ijms, 2023, doi:10.3390/ijms24087487_

Round 1

Reviewer 1 Report

In this review by Montserrat Fernández-Guarino et al., the authors aim to summarize the role of physical therapies in wound healing and scarring. Wound healing is a complex process involving multiple biological pathways and mechanisms. The failure of wound healing, leading to chronic wounds or excessive scarring, can significantly impact patients' quality of life. The authors discuss the different stages of wound healing and explore the role of physical therapies such as laser, low-level laser light therapy, photodynamic therapy, and electrical stimulation in promoting wound healing.

Nevertheless, there are 2 minor suggestions for improvement are proposed for this review:

1.       The introductory section discussing wound induction appears to be redundant. The authors are encouraged to condense Section 2 for enhanced clarity and conciseness.

2.       Can authors include and summarize additional information from clinical trials investigating the efficacy of physical therapy interventions for wound healing, thereby providing a more robust evidence base for their conclusions.

Author Response

ANSWER TO REWIEVER ONE

Firstly, I would like to thank the reviewer for his/her time in revising the manuscript, I believe it is a great effort as is time consuming. Also, it is a great guide for me, to improve my writing and my view of an article. So, I am grateful for that.

I have copy below the comments of the reviewer 1, and afterwards, I have added my comments and the descriptions of the changes. The changes and the new reference order are highlighted in the new submitted manuscript.

For a better comprehension I am showing the eliminated text in red color and the changed final text in green.

In this review by Montserrat Fernández-Guarino et al., the authors aim to summarize the role of physical therapies in wound healing and scarring. Wound healing is a complex process involving multiple biological pathways and mechanisms. The failure of wound healing, leading to chronic wounds or excessive scarring, can significantly impact patients' quality of life. The authors discuss the different stages of wound healing and explore the role of physical therapies such as laser, low-level laser light therapy, photodynamic therapy, and electrical stimulation in promoting wound healing.

Nevertheless, there are 2 minor suggestions for improvement are proposed for this review:

1.       The introductory section discussing wound induction appears to be redundant. The authors are encouraged to condense Section 2 for enhanced clarity and conciseness.

-In section 2.1 (Line 81), this paragraph is erased, as in the next sections the types of chronic wounds are described:

“The estimated cost per wound depends on the treatment applied and ranges from 3.927 to 9.358 dollars, being graft treatment the most expensive. Most of these CW comprised: diabetic foot ulcers (DU), venous leg ulcers (VU) and pressure ulcers (PU) [1].”

-Section 2.2. and subsections, are describing the phases of wound healing and the biological signals and effectors included. From my point of view, those sections are necessary for understanding the article, so I have not changed them.

-I eliminate section 2.2.4, as maybe it is not as important to a comprehensive reading of the manuscript. Coherently, I eliminate Table 2 and numbered again the Tables.

2.2.4. Scars

Remarkably WH in a fetus occurs without scarring, and “restitutio ad integrum” produced normal skin with neovascularization and appendages. Therefore, the study of the process of scarring in the fetus has been focused on the scientist trying to find out the differences and mimicking them [23]. In the next table, the differences between fetal and adult scarring are summarized (Table 2). Fetal WH exhibits lower inflammation and higher matrix production with more collagen type III and hyaluronic acid [2,23,24].

Table 2. Differences between fetal and adult skin scarring.

Process

Fetal skin

Adult skin

Inflammatory cells

Fewer

More

Cytokines profile

TGFb3

IL1, IL10 (anti-inflammatory)

TGFb1, TGFb2

IL6, IL8 (proinflammatory)

Extracellular matrix

Higher production by fibroblasts

Collagen type I > type III

Collagen type III< type I

Lower quantity of hyaluronic acid

Absence of myofibroblasts

Myofibroblasts

TGF: transforming growth factor; IL: interleukin.

-In section 2.3 I proceed to eliminate some text, as the second and the third paragraph talking about the concomitant factors in the management of chronic wounds, as the revision is focused on physical therapies, is this one:

“CW is more prevalent in lower economic classes [1]. Clinical management is focused on the prevention and improvement of the concomitant factors affecting WH, however, novel approaches are needed as many CW remains refractory.

Infected CW exhibits different signs from infected AW; therefore, the typical signs of edema, erythema or pain could not be present [13]. In CW both biofilm, critical colonization, or infection could produce exudates, bleeding, new areas affected, erythema and smells [25]. It must be outlined that the best way to assess the presence of infection in a CW is by the culture of a cutaneous biopsy and not by direct samples, thus, it is not easy to diagnose [26].

Infected CW exhibits different signs from infected AW; therefore, the typical signs of edema, erythema or pain could not be present [13]. In CW both biofilm, critical colonization, or infection could produce exudates, bleeding, new areas affected, erythema and smells [25]. It must be outlined that the best way to assess the presence of infection in a CW is by the culture of a cutaneous biopsy and not by direct samples, thus, it is not easy to diagnose [26]. “

-In section 2.4 I made multiple changes, eliminating redundant information and keeping only the information which, I believe it is necessary for understanding the next sections of the manuscript.

The paragraphs finally appear like this (the changes are highlighted in the text):

“WH and scarring is a complex process with multiple influencing and interacting factors. Besides, some of those factors are not under the control of the dermatologist, such as age, vascular abnormalities, comorbidities, malnutrition, or smoking.[1] The management is challenging, and multiple approaches and visits are needed with the implication of different health care workers [1] and arisen important indirect costs.

All CW should be treated according to the TIME principles: tissue debridement, infection control, moisture balance and edges of the wound [13]. Debridement is the first step in the treatment of a CW, it must be done weekly and increase the speed of healing by over 72% [31].

Biofilm is presented in the extracellular matrix and is considered the responsive of 80% of the infections in CW [32]. Biofilm cannot be shown with eyes, and different techniques to assess its presence are being developed apart from a cutaneous biopsy. Anyway, biofilm must be removed because maintains the CW in the inflammatory stage [33]. The risk of infection is usually controlled by topical antibiotics or silver dressing or with other topical components.

The wound should not be exposed to air, and if the skin appears dry, moisturizing should be added to the dressing. On the other hand, if excessive drainage is presented, it needs to be clean and dry. The wound edge, in case of overgrowth, must be excised for epithelization [34].

.

Table 2. describes the local cellular response alterations underlying a CW. CW are characterized by excessive inflammation, a decrease of growth factors secretion, disbalance in the proteolytic enzymes and cellular senescence which perpetuates the wound unhealed [35]. Therefore, a high number of mast cells, neutrophils and dendritic cells are founded in CW with an increase of pro-inflammatory cytokines and proteases (see Figure 1). These inflammatory cells not only perpetuate the wound but also increase susceptibility to infections [22]. The alteration of the expression and activation of MMPs is strongly associated with CW, damaging the granulation tissue, and producing exudates in the wound [35]. Cells implicated in the remodeling and re-epithelization are dysfunctional too. Fibroblasts are senescent and the excess of wound proteases (MMPs, elastase, cathepsin G, and urokinase-type plasminogen activator (uPA) activities degrade the extracellular matrix, the growth factors (VEGF, TGF-beta) and cytokines (TNF-alfa) [22]. Keratinocytes hyper proliferate in the edge of the wound, so hyperkeratosis appears and the subsequent failure to close. “

-In section 2.5, I eliminate de first sentence as it maybe sound redundant.

2.       Can authors include and summarize additional information from clinical trials investigating the efficacy of physical therapy interventions for wound healing, thereby providing a more robust evidence base for their conclusions.

Yes, and I believe it is a very good advice. I have come across the literature and add this paragraph at the end of the revision in section 6:

5. Summarize of clinical trials of physical therapies in wound healing

Finally, it is important to take in consideration that there are few clinical trials published about interventions with PT in WH. When the term “Physical therapies skin wounds” is introduced in “Pubmed” and filters added as, “last 10 years”, “clinical trials” a total of 66 results appeared. After been selected the, only 8 works were focused on skin and only 6 in CW (Figure S1).

PT are not included in clinical Guidelines of management of CW, and they are used as adjuvant therapies[1]. Polak treated with electrical stimulation (ES) areas of pressure injuries in patient with neurological damage. A total of 43 patients were divided in three groups, anodal, cathodal or placebo. A diminution of proinflammatory blood cytokines was found in the patients treated in correlation with an improvement of the clinical peripheric inflammation of the wound and a significant reduction in the size of the wound was assessed, in comparison to the placebo group. The same therapy was used for other authors[3] to explore de effect in CW in combination with silver dressing. Ten patients were treated only in one wound and the rest were used as controls, finding significant differences. ES has been also proved to alleviate pain in CW, apart from accelerate healing[4]. In a study in 10 patients compared with placebo in other 10, ES improved in DU the blood levels of VEGF and NO [5].

In an interesting study, ultrasound (US) and electrical stimulation (ES) were compared in the treatment of PU. Both treatments improved WH in 27 patients treated without significant differences between them [6]. Another comparative study was carried out by Polak in 77 patients with PU using standard wound care, US and ES. Patients treated with PT had a significant decrease of the ulcer area compared with placebo without differences between US and ES[7].The main limitation of all those clinical trials is control others factor that could influence in the healing of the wound.

An also these references:

  1. Gupta, S., et al., Management of Chronic Wounds: Diagnosis, Preparation, Treatment, and Follow-up. Wounds, 2017. 29(9): p. S19-S36.
  2. Polak, A., et al., A Randomized, Controlled Clinical Study to Assess the Effect of Anodal and Cathodal Electrical Stimulation on Periwound Skin Blood Flow and Pressure Ulcer Size Reduction in Persons with Neurological Injuries. Ostomy Wound Manage, 2018. 64(2): p. 10-29.
  3. Zhou, K., et al., Silver-Collagen Dressing and High-voltage, Pulsed-current Therapy for the Treatment of Chronic Full-thickness Wounds: A Case Series. Ostomy Wound Manage, 2016. 62(3): p. 36-44.
  4. Fraccalvieri, M., et al., Electrical stimulation for difficult wounds: only an alternative procedure? Int Wound J, 2015. 12(6): p. 669-73.
  5. Mohajeri-Tehrani, M.R., et al., Effect of low-intensity direct current on expression of vascular endothelial growth factor and nitric oxide in diabetic foot ulcers. J Rehabil Res Dev, 2014. 51(5): p. 815-24.
  6. Bora Karsli, P., et al., High-Voltage Electrical Stimulation Versus Ultrasound in the Treatment of Pressure Ulcers. Adv Skin Wound Care, 2017. 30(12): p. 565-570.
  7. Polak, A., et al., Reduction of pressure ulcer size with high-voltage pulsed current and high-frequency ultrasound: a randomised trial. J Wound Care, 2016. 25(12): p. 742-754.

I am sending the changes highlighted and the clean version.

Your sincerely,

Montserrat Fernandez-Guarino, PhD

Reviewer 2 Report

The scientific paper "The Role of Physical Therapies in Wound Healing and Assisted Scarring” aimed to summarize the role of physical therapies as complementary treatments in Wound Healing and scarring.

The manuscript is well written. It has scientific merit. The subject is of great clinical importance.

I can make the following considerations:

1)       References are not in the IJMS MDPI journal standard. Please adjust;

2)       Merge the first 2 paragraphs of the introduction into a single paragraph;

3)       Increase the size of the letters that are inside figure 1 as they are not visible;

4)       Increment (increase) the subtitle 2.3.2 (Pressure ulcer). Due to its importance, it deserves to be complemented;

Author Response

ANSWER TO REWIEVER TWO

Firstly, I would like to thank the reviewer for his/her time in revising the manuscript, I believe it is a great effort as is time consuming. Also, it is a great guide for me, to improve my writing and my view of an article. So, I am grateful for that.

I have copy below the comments of the reviewer 1, and afterwards, I have added my comments and the descriptions of the changes. The changes and the new reference order are highlighted in the new submitted manuscript.

For a better comprehension I am showing the eliminated text in red color and the changed final text in green.

The scientific paper "The Role of Physical Therapies in Wound Healing and Assisted Scarring” aimed to summarize the role of physical therapies as complementary treatments in Wound Healing and scarring. 

The manuscript is well written. It has scientific merit. The subject is of great clinical importance.

I can make the following considerations:

  • References are not in the IJMS MDPI journal standard. Please adjust;

Ok I will change them.

  • Merge the first 2 paragraphs of the introduction into a single paragraph;

Ok, I proceed to change it.

  • Increase the size of the letters that are inside figure 1 as they are not visible;

I increase the size onf the letter in the figure and change it.

  • Increment (increase) the subtitle 2.3.2 (Pressure ulcer). Due to its importance, it deserves to be complemented;

Ok, I add the next information.

“PU appears on areas under pressure usually over a bony prominence such as the sacrum or the heels. The pression on the vessels decrease irrigation of the skin resulting in and initial dermatitis which in case of been prolonged leads to a loss of tissue. The cause is multifactorial as immobilization in bed, nutrition alteration, systemic diseases, or elderly [1]. PU varies in severity, and are classified in four different stages, being the IV the more severe, which implies the loss of the full thickness of the skin. In those cases, the management of the PU should be surgical, and PT would not play a role”

I am sending the changes highlighted and the clean version.

Your sincerely,

Montserrat Fernandez-Guarino, PhD
